# High and Low Dosage of Vancomycin in Polymethylmethacrylate Cements: Efficacy and Mechanical Properties

**DOI:** 10.3390/antibiotics13090818

**Published:** 2024-08-28

**Authors:** Jeffrey W. Kwong, Michael Abramowicz, Klaus Dieter Kühn, Christian Foelsch, Erik N. Hansen

**Affiliations:** 1Department of Orthopaedic Surgery, University of California San Francisco, San Francisco, CA 94143, USA; jeffrey.kwong@ucsf.edu (J.W.K.); erik.hansen@ucsf.edu (E.N.H.); 2KABEG, LKH Villach, 9500 Villach, Austria; michael.abramowicz@kabeg.at; 3Department of Orthopaedics and Trauma, Medical University of Graz, 8036 Graz, Austria; 4Department of Orthopaedics and Orthopaedic Surgery, University Hospital Gießen and Marburg (UKGM), Justus-Liebig-University, Klinikstraße 33, 35392 Gießen, Germany; christian.foelsch@ortho.med.uni-giessen.de

**Keywords:** two-stage revision, periprosthetic joint infection, vancomycin, PMMA spacer, Copal G + C, Copal G + V, Copal Exchange G, Cemex Spacer K

## Abstract

Introduction: Prosthetic joint infections (PJIs) are difficult to treat and represent a significant burden to the healthcare system. Two-stage revision surgery with placement of an antibiotic-loaded cement spacer is currently the gold standard for treatment in the United States for late-onset infections. We evaluate the efficacy of varying doses of vancomycin added to antibiotic-containing acrylic cement spacers and discuss the biomechanical and antimicrobial properties of using high versus low doses of vancomycin in cement spacers in the hip and knee. Materials and Methods: Commercially available Copal cement containing either gentamicin and clindamycin (G + C) or gentamicin and vancomycin (G + V) was prepared with the manual addition of low (2 g) and high (6 g) doses of vancomycin. In vitro mechanical testing was then carried out according to ISO 5833 and DIN 53435, as well as inhibition zone assays against common PJI pathogens. Additionally, inhibition zone assays were conducted on two commercially available prefabricated spacers containing gentamicin: Copal Exchange G and Cemex Spacer-K. Results: In biomechanical testing, Copal G + V with the addition of 6 g of vancomycin failed to meet the ISO standard. Copal G + C and Copal G + V with low and high dosages of vancomycin were all effective against the tested pathogens and displayed constant efficacy for a duration of 42 days. High doses of vancomycin showed significantly lower mechanical stability. Moreover, Copal Exchange G showed significantly larger inhibition zones across 42 days. Discussion: While higher concentrations of vancomycin appear to improve the antimicrobial efficacy of cement, they also reduce its mechanical stability. Despite its smoother surface, the Copal Exchange G spacer exhibits large inhibition zones after 1 day and maintains consistently large inhibition zones over 6 weeks. Thus, it may be preferred for use in two-stage revision surgery. Conclusion: Copal Exchange G is more effective than Cemex Spacer K against *S. aureus* and *E. coli*. The manual addition of vancomycin to cement containing double antibiotics is very effective. The influence on ISO compression is low, the ISO bending modulus is increased, and ISO bending, DIN bending, and DIN impact, are reduced.

## 1. Introduction

Total hip arthroplasty (THA) and total knee arthroplasty (TKA) are among the most common procedures performed worldwide given their demonstrated track record of relieving pain and improving function [1,2,3,4]. Although revision surgery is relatively rare, the number of revisions is increasing and the number of primary arthroplasties continues to grow [5,6,7,8]. Prosthetic joint infections (PJIs) are expected to contribute significantly to the increase in the number of revision hip arthroplasties performed [9], and PJIs are currently the leading cause of TKA revisions [10,11,12,13]. By 2030, it is estimated that there will be more than 250,000 cases of PJI per year [12], which are projected to impose a USD 1.85 billion economic burden on the healthcare system [14]. Consequently, effective treatment of PJIs is critical.

In the United States, two-stage revision arthroplasty remains the gold standard for the treatment of late-onset PJI [15,16,17]. In the first stage, a thorough surgical debridement is performed, including complete removal of existing implants given the concern for biofilm formation, followed by implantation of an antibiotic spacer, typically made of polymethylmethacrylate (PMMA) cement. These antibiotic-laden spacers serve several purposes: they preserve soft tissue tension, maintain functionality, and deliver local antibiotics to the infected tissue bed [15,18,19]. After 4–6 weeks of parenteral antibiotic therapy, if there are no further signs of infection, patients undergo a second-stage surgery to remove the spacer and replace it with a new, definitive implant [15,20,21].

The amount and type of antibiotic added to the PMMA spacer during the first stage of surgery remain debated [20]. The chosen antibiotic must be effective at treating the causative organism, have minimal systemic side effects, and not adversely affect the biomechanics of the PMMA cement. Here, we focus on vancomycin and summarize the risks and benefits of its addition to PMMA spacers for the treatment of PJI.

Vancomycin is a water-soluble, thermostable glycopeptide that works by inhibiting bacterial cell wall biosynthesis [22,23,24,25]. It is used in the treatment of Gram-positive organisms, including infections from methicillin-resistant *Staphylococcus aureus* (*MRSA*) and methicillin-resistant *Staphylococcus epidermidis* (*MRSE*) [20]. Given that these bacteria are among the most frequent causative organisms of PJI of the hip and knee [26,27], it is not surprising that vancomycin is often chosen to be added to PMMA cement spacers during two-stage revision arthroplasty.

The amount of vancomycin added to cement spacers can be divided into high or low doses. Although definitions vary somewhat, high-dose vancomycin has generally been defined as the addition of more than 3.6 g of vancomycin per 40 g bag of PMMA cement, while a low dose has been defined as 1 g or less per 40 g bag of cement [18,28,29,30]. Doses between 1 to 3.6 g of vancomycin per 40 g bag of cement have not been clearly classified but may be appropriately referred to as intermediate doses.

Commercially available formulations of PMMA may or may not include antibiotics in the cement powder. However, those that do contain only low doses of vancomycin [31]. Multiple authors have commented that 1 g of vancomycin or less per 40 g bag of cement is sufficient only for prophylaxis and is not appropriate for the treatment of a known infection [18,20,31]. In the setting of treatment for PJI, 4 g of vancomycin per 40 g bag of cement is recommended, and 2 g of vancomycin per 40 g bag of cement is considered the minimum [28].

Recent studies suggest that higher doses of vancomycin are associated with higher rates of treatment success for PJI. Corró et al. demonstrated that increasing the dose of antibiotic from 1 g each of vancomycin and gentamicin to 5 g of each antibiotic per 40 g bag of PMMA was associated with a greater probability of eradication of the infection at 2-year follow-up [32]. Additionally, Warwick et al. reported that vancomycin doses of at least 2 g per 40 g of cement and a total antibiotic dose of at least 3.6 g per 40 g of cement were associated with lower odds of infectious failure [33]. Consequently, emerging literature appears to support the use of high-dose vancomycin in cement spacers.

In the current study, we assess the efficacy of varying doses of vancomycin added to commercially available antibiotic-loaded acrylic cement, as well as two prefabricated gentamicin-containing spacers. We discuss the biomechanical and antimicrobial properties of using high versus low doses of vancomycin in cement spacers in the hip and knee.

## 2. Results

### 2.1. ISO Compression Strength

All tested cement combinations fulfilled the requirements for ISO compression. The standard deviation between the tested cements was low, without any statistical outliers. The addition of high dosages of vancomycin increased the ISO compression strength. The control Copal G + C showed greater compression strength compared to the control Copal G + V (Figure 1, green bars). Copal G + C with 2 g vancomycin showed the lowest strength in compression (Figure 1).

### 2.2. ISO Bending Modulus

The results of the bending modulus were all within a similar range, without significant statistical outliers. Adding high concentrations of vancomycin to both Copal cements increased the average bending modulus (Figure 2).

### 2.3. ISO Bending Strength

After adding vancomycin to both Copal cements, the bending strength was reduced. There was no significant difference between Copal G + C and Copal G + V. Copal G + C plus 6 g vancomycin resulted in a bending strength just above the ISO minimum (50.9 MPA), while Copal G + V with the same vancomycin concentration failed to meet the minimum (43.3 MPa) (Figure 3).

### 2.4. DIN Bending Strength

In the DIN bending tests, only Copal G + C without added vancomycin met the minimum bending strength requirements. All other tested cements had bending strengths below the DIN minimum. Higher doses of vancomycin resulted in lower bending strength (Figure 4).

### 2.5. DIN Impact Strength

The DIN impact strength was reduced after adding vancomycin to the tested cement. The higher the dose of vancomycin added to the cement, the lower the impact strength. The two Copal cements behaved similarly (Figure 5).

### 2.6. Efficacy against Staphylococcus aureus

All cements were effective against *S. aureus* over the tested period of 42 d. All inhibition zones remained high over the entire test period. Copal G + C with vancomycin, which contained three antibiotics, had a wider inhibition zone compared to Copal G + V with added vancomycin, which contained two antibiotics. There was a small difference observed between low and high dosages of vancomycin for both Copal groups (Figure 6).

### 2.7. Efficacy against E. faecalis

All cements were effective against *E. faecalis* over the tested period of 42 d. All inhibition zones remained high over the entire test period. The inhibition zones of Copal G + C with vancomycin, which contained three antibiotics, and Copal G + V with vancomycin, which contained two antibiotics, were similar. There was only a small difference observed between low and high dosages of vancomycin for both Copal groups. Higher vancomycin concentrations had larger zones of inhibition (Figure 7).

### 2.8. Efficacy against MRSA

All tested cements were effective against *MRSA* over the tested period of 42 d. All inhibition zones remained high over the entire test period. The was no difference between Copal G + C with vancomycin, which contained three antibiotics, and Copal G + V with vancomycin, which contained two antibiotics. There was a small difference observed between low and high dosages of vancomycin for the two Copal groups. Higher vancomycin concentrations had larger zones of inhibition (Figure 8).

### 2.9. Prefabricated Spacers with Gentamicin

Copal Exchange G and Cemex Spacer K were compared according to their antimicrobial efficacy by using inhibition zone tests against predetermined bacteria. The diameter of the inhibition zones was measured in mm. Both spacers were tested against the following bacteria: *S. aureus*, *MRSA*, *E. coli*, and *E. faecalis*.

No efficacy was observed for both tested spacers against *E. faecalis* and *MRSA*.

Both tested spacers were effective against *S. aureus* over a period of 42 days. In all cases, Copal Exchange G had greater inhibition zones in mm and eluted cumulatively more gentamicin than Cemex Spacer K over time. The largest inhibition zone was observed after 1 day for Copal Exchange G (Figure 9).

Both tested spacers were also effective against *E. coli*. Copal Exchange G was effective over the course of 42 days whereas Cemex Spacer K was no longer effective after 14 days. Additionally, Copal Exchange G eluted cumulatively more gentamicin than Cemex Spacer K over time. The largest inhibition zone was observed after 1 day and 7 days for Copal Exchange G (Figure 10).

## 3. Discussion

### 3.1. Microbiology

Chronic infections and infections due to virulent bacteria often necessitate two-stage revisions of prosthetic joint infections [34,35,36,37,38,39,40]. Failure of treatment after two-stage revision is often related to staphylococci or new bacteria recovered in the re-implantation procedure, suggesting the initial infection had already been polymicrobial [41,42]. Treatment of culture-negative PJI and low-grade infections with difficult-to-treat bacteria require high local antibiotic concentrations [43]. Previous work suggests that treatment with local antibiotics does not result in specific antimicrobial resistance [44].

Polymicrobial infections often present with a shorter time between implantation and onset of symptoms. While some authors have claimed the rate of treatment success for PJIs with biofilm-forming bacteria is not lower than PJIs due to other bacteria [43,45,46,47], others have reported lower rates of eradication for infections with rifampicin-resistant bacteria, even after two-stage revision [48]. Indeed, there appears to be a growing number of failed two-stage revisions for PJIs [49,50,51,52,53,54].

Biofilms pose a particular challenge to the treatment of PJI. Irreversible attachment of bacteria on surfaces and local tissues begins within one minute and increases 100 times after 10 min [55]. While bacteria-that produce biofilms are up to 1000 times more resistant to antimicrobial agents, mature biofilms need 4 weeks to develop [37,56,57], so early treatment of PJI is key. Some evidence suggests local antibiotic delivery with antibiotic cement might increase the minimal infecting dose of *S. aureus* by more than 100,000-fold [58]. Additionally, the use of various antibiotic carriers may be important to deliver local antibiotics [55]. Winkler et al. also found that the addition of 2 g of vancomycin might allow a shorter interval of spacer treatment [59], and the use of dual-loaded antibiotic cement appears to improve eradication rates in infected TKAs [59,60,61,62]. Consequently, the high local antibiotic availability released from cement spacers might be useful to improve the eradication of bacteria and shorten the period of spacer treatment [53,63,64,65,66].

Although the amount of antibiotic added to a cement spacer clearly has a significant impact on the local tissue availability of the antibiotic, other factors, including the viscosity and type of cement, the mixing technique, and the specific combination of antibiotics added can all affect the amount and pharmacokinetics of the antibiotic delivered to the surrounding tissues [67,68,69].

In vitro studies have suggested that the elution of vancomycin from an antibiotic spacer is not linear. Rather, there is a large burst of antibiotic initially, followed by an asymptotic taper [70]. Consistent with this model, the amount of locally released antibiotic has been found to be greatest in the first 48 h [71], and the greater the amount of antibiotic used in the cement, the longer the antibiotic remains detectable [72]. Chang et al., for example, reported it took 2 days for local vancomycin to reach undetectable levels when 1 g was added per 40 g of PMMA, while it took 21 days when 4 g were added [70]. With 8 g of vancomycin, the antibiotic was still detectable in the eluent at 60 days, albeit at very low levels. The findings of the current study appear to support these conclusions. With antibiotic spacers that contained gentamicin only, the diameter of the inhibition zone appeared to decrease with incubation time. However, when high doses of vancomycin were added to cement, the antimicrobial activity of the cement was sustained across 42 days. Adding high doses of vancomycin to commercially available cement mixtures may, therefore, be one strategy to maintain the local antibiotic activity of cement spacers for extended periods of time.

These findings are confirmed by several in vivo studies. Using 4 g of vancomycin per 40 g of PMMA for patients with PJI of the hip, Hsieh et al. examined the joint fluid at the time of spacer explantation [22]. At a mean of 107 days since the index surgery, vancomycin was still detected at levels above the minimum inhibitory concentration, and they found that the joint fluid remained bioactive against *S. aureus*. Importantly, they reported serum levels of vancomycin were low at all tested time points, and all patients had serum levels below the threshold of detection by 72 h after the first-stage surgery. Thus, the addition of vancomycin to PMMA appears to be an effective method of achieving high doses of local antibiotics with minimal systemic availability. While low doses of vancomycin (≤1 g/40 g of cement) may not result in the release of the antibiotic for the full length of time between the first- and second-stage surgeries, intermediate doses (2–3.6 g/40 g of cement) are likely sufficient, and high doses (≥3.6 g/40 g of cement) appear to provide sustained release of the antibiotic in the order of months.

Additionally, the porosity of the cement plays a critical role in the elution of vancomycin from the spacer. More porous cement is better able to release the antibiotic into the surrounding soft tissue, and, to this end, some have advocated for the addition of dextran to the cement mixture to further increase porosity [73]. Similarly, studies have found that adding vancomycin after initially mixing the powdered cement with the liquid phase for 30 s creates more pores for cement elution and can result in a greater amount of locally available vancomycin [74]. Hand mixing the antibiotic with the cement is generally considered favorable, as this technique leaves large crystals of antibiotic that increase porosity; however, it can also result in a non-homogenous distribution of the antibiotic which introduces some variability into the amount of eluted antibiotic [18,30] Mixing the cement at atmospheric pressure, without a vacuum, can also increase porosity [18,75,76,77]. Using a high-viscosity cement such as Palacos (Heraeus Medical, GmbH, Wehrheim, Germany) has also been associated with greater elution and a higher likelihood of successful eradication of PJI [33,77,78,79,80,81,82,83].

The combination of antibiotics used in the cement spacer has been reported to affect the elution of vancomycin. Penner et al., for example, reported that the elution of vancomycin and tobramycin was greater when the antibiotics were used together than when vancomycin was added in isolation to PMMA cement [84]. This effect was theorized to be due to the additional volume from the second antibiotic, which increases the porosity of the cement as it dissolves, similar to the effect of adding dextran. Others have reported that the addition of cefazolin similarly potentiates the elution of vancomycin compared to vancomycin alone [71], suggesting the volume effect of the second antibiotic may be most critical, rather than the specific choice of antibiotic. However, Slane et al. found that cement with the largest amount of loaded antibiotic did not have the best elution kinetics, and they found that a combination of 3 g of tobramycin and 2 g of vancomycin was the most optimal [85]. This result may suggest a more complex interplay of the physical interactions between the antibiotic and the cement and between the antibiotics themselves [67,86].

There are some concerns regarding systemic toxicity, such as nephrotoxicity, due to the uncontrolled elution of antibiotic-containing PMMA spacers [87,88,89]. Nevertheless, multiple studies have recently called this paradigm into question [19,22,90,91]. Thus, systemic toxicity due to the antibiotics released from cement spacers seems unlikely, and the clinical practice guidelines from the Infectious Disease Society of America agree that systemic toxicity from antibiotics used in cement is exceedingly rare [21].

### 3.2. Biomechanics

The amount of vancomycin used can alter the mechanical integrity of cement. In vitro biomechanical studies have confirmed that increasing the amount of antibiotics can weaken the PMMA and decrease the compressive, diametral, and flexural strength to levels below the ISO standard [85,92]. Of note, in the current study, high dosages of vancomycin increased the ISO compression strength of Copal cement, although the bending strength and impact strength were decreased. The discrepant results observed with compressive strength may be due to a difference in the mechanical properties of the specific type of cement tested, which was Copal in the results reported here, compared to Palacos [85] and Simplex [92] when reported elsewhere. Alternatively, this difference may be attributed to dissimilar testing protocols used across studies. Nonetheless, it is important to exercise caution when using cement with manually added antibiotics, as the mechanical properties of the resulting mixture are often compromised.

Although the specific choice and combination of antibiotics appears to have some effect on these parameters, the total mass of antibiotics used appears to be the most important [56,70,77,85]. For daptomycin-containing PMMA cements, a significant reduction in bending strength with little influence on compression and bending modulus was observed by Humez et al. [93]. Similar results were reported by Krampitz et al. after adding voriconazole to PMMA [94]. This decrease in strength may be due to an increase in porosity from the addition of large amounts of antibiotics [18,95]. On the other hand, a significant reduction in bending strength combined with a slight increase in compression and bending modulus may be a consequence of an increase in the hydrophilicity of the cement matrix [77]. Consequently, cement with ≥4.5 g of antibiotic should not be used for implant fixation, as the compressive strength of cement with such large amounts of antibiotic has been found to be below the minimum standard [18,92,96]. Furthermore, while high-dose vancomycin is preferred for creating a spacer in the first stage of a two-stage exchange, at most 1–2 g of antibiotic powder should be used during the re-implantation stage for fixation [30], when the implant is meant to stay permanently and the integrity of the cement is necessary to create a strong and lasting bond between the bone and the implant.

Along the same lines, all factors that increase the porosity of the cement can weaken its strength [77]. This includes the use of dextran [73], mixing at atmospheric pressure [76], and adding antibiotics to the cement after combining the powder and liquid phases of the cement [74].

Furthermore, the strength of the antibiotic cement changes with time as the antibiotic elutes from the PMMA. Amin et al. demonstrated significant decreases in the compressive strength of PMMA with 5 g of vancomycin after 6 weeks of incubation in vitro [74]. In one case, although the compressive strength of the antibiotic cement was above the ISO minimum at the outset, it dropped below this minimum at the end of incubation. Paz et al. similarly confirmed that when cefazolin and/or vancomycin were added to PMMA, the cement demonstrated a reduction in compression strength and bending strength after 1 month of incubation, no matter the dose that was used [71]. Only cement with no antibiotic added demonstrated retention of its mechanical properties. The reduced strength of antibiotic-loaded cement with time may be explained by the voids and microcracks formed from the elution of the antibiotics, which Paz et al. assessed using scanning electron microscopy [71]. Given the compromised biomechanical properties of antibiotic-loaded cement, restricted weight bearing may be considered between the first- and second-stage surgeries. Moreover, surgeons should exercise caution if the second-stage re-implantation surgery is delayed, as these data indicate the strength of the spacer declines with time, and catastrophic failure may result from prolonged retention of the spacer [97].

## 4. Methodology

Commercially available Copal PMMA cement (Heraeus Medical GmbH, Wehrheim, Germany) containing either gentamicin and clindamycin (1 g G + 1 g C) or gentamicin and vancomycin (0.5 g G + 2 g V) was used. Copal G + C is characterized by the following composition: 42.7 g of Copal G + C powder contains 1.0 g gentamicin (as gentamicin sulfate) and 1.0 g clindamycin (as clindamycin hydrochloride). Other ingredients include: Poly(methyl methacrylate/methacrylate), Zirconium dioxide, Benzoyl peroxide, and colorant E141. A 20 mL monomer liquid contains: methyl methacrylate, dimethyl-p-toluidine, hydroquinone, and colorant E 141 [98]. Copal G + V is composed as follows: 43.0 g Copal G + V cement powder contains 0.5 g gentamicin (in the form of gentamicin sulfate) and 2.0 g vancomycin (in the form of vancomycin hydrochloride). Other ingredients include: poly(methyl methacrylate/methacrylate), zirconium dioxide, benzoyl peroxide, and colorant E 141. A 20 mL monomer liquid contains methyl methacrylate, dimethyl-p-toluidine, hydroquinone, and colorant E 141. Further material characterization of Copal has been previously reported [77,99,100,101].

Additionally, we used two commercially available prefabricated spacers containing gentamicin, the Copal Exchange G knee spacer, size S (Heraeus Medical GmbH, Wehrheim, Germany), and the Cemex Spacer K knee spacer, size S (Tecres S.p.A, Sommacampagna, Italy). All cements were prepared according to the instructions of the manufacturer [98,101].

To manufacture the cement blocks, vancomycin powder with a 90.2% antibiotic concentration was used. Thus, 2.21 g and 6.65 g of antibiotic powder were added to reach 2 g and 6 g of pure vancomycin, respectively. Each vancomycin dose was then combined with 1 bag of Copal G + C (42.7 g powder containing 1 g of gentamicin and 1 g of clindamycin) or 1 bag of Copal G + V (43 g powder containing 0.5 g Gentamicin and 2 g Vancomycin). The cement was mixed according to the instructions of the manufacturer. Afterwards, the cement dough was cast into stainless steel molds. A total of 3 different molds were used for the various mechanical and microbiological tests.

For microbiological tests, the following strains of bacteria were used: Methicillin-sensitive *S. aureus* ATCC 29213, *MRSA* ATCC 43300, *E. faecalis* ATCC 29212, and *E. coli* ATCC 25922 (see Table 1).

### 4.1. Mechanical Tests

#### 4.1.1. ISO 5833 Compressive Strength

The International Organization for Standardization (ISO) compressive strength test was used to determine the pressure or compressive force that is needed until the cement loses its stability and breaks. All cement specimens (height: 12 mm ± 1, diameter: 6 mm ± 0.1, tested in replicates of 12) were placed in the middle of a test machine capable of applying and measuring compressive force (Zwick/Roell, Ulm, Germany) and running the testXpert II Zwick/Roell software (https://www.zwickroell.com/accessories/testxpert-testing-software/, accessed on 31 December 2021). The machine applied increasing force until the specimen fractured, or until the 2% offset load or upper-yield-point load was reached. At that point, the internal pressure was measured in MPa. To comply with the standards of the ISO 5833 [102], specimens had to reach a minimum internal pressure of 70 MPa on average.

#### 4.1.2. ISO 5833 Bending Modulus and Bending Strength

For each cement combination, rectangular cement bodies (75 × 10 × 3.3 mm) were used and tested in replicates of 6. After extracting the cement bodies from the stainless steel molds, they were placed in a four-point test rig (Zwick/Roell, Ulm, Germany) running the testXpert II Zwick/Roell software. Care was taken to place the test specimens as centrally on the device as possible. The machine applied increasing force on the test specimen while measuring its deflection until the specimen broke. The software then calculated the bending and strength modulus of each cement combination in MPa. To comply with the standards of ISO 5833 [102], specimens had to reach a minimum bending modulus of 1800 MPa and a bending strength of 50 MPa.

#### 4.1.3. DIN Bending and DIN Impact Strength according to DIN 53435

The bending and impact strength of each cement combination was tested against the Deutsches Institut für Normung (DIN) 53435 standard [103]. For each cement combination, rectangular cement bodies (15 × 10 × 3.3 mm) were used and tested in replicates of 8. The specimens were placed in the DYNSTAT bending test apparatus (Zwick/Roell, Ulm, Germany). The apparatus began to rotate at 100°/min, applying a bending moment of 400 Ncm on the specimen. Once the specimen broke, the machine was stopped and the bending moment of the test body at the breaking point was recorded in Ncm. To determine the impact strength, specimens were placed in the DYNSTAT strength apparatus (Zwick/Roell, Ulm, Germany) and the pendulum was placed in its starting position. Once the pendulum was released, it collided with the test body with an impact energy of 0.5 J. The required impact energy (KJ/m^2^) to break the test body was then recorded. To comply with the standards of DIN 53435 [103], the specimens had to reach a minimum bending strength of 65 MPa. Impact strength results are compared against reference data.

For all mechanical tests, the number of samples and statistical evaluation were carried out in accordance with the standards specified by ISO 5833 and DIN 53435 [102,103].

### 4.2. Spacers

Both Copal Exchange G and Cemex Spacer K contained roughly equal concentrations of gentamicin. To properly compare both spacers according to their antibiotic release and efficacy and to simulate in vivo conditions that occur within the joint, the concave inner, non-articulating surface of each spacer was covered with antibiotic-free bone cement (Palacos R, Heraeus Medical GmbH, Wehrheim, Germany, see Figure 11). Each spacer was tested in triplicate. Once the cement had hardened, the spacers were examined for their antimicrobial activity against bacteria. Of note, the Copal Exchange G spacer had a smooth surface and calcium carbonate (CaCO_3_) as the contrast agent for X-rays, while the Cemex Spacer K had a rough, porous surface and barium sulfate (BaSO_4_) as the contrast agent.

### 4.3. Microbiological Tests

All test specimens (25 × 10 mm) and spacers (size S) were tested to examine the antibacterial efficacy via an inhibition zone assay. Table 1 illustrates which test specimens and spacers were used against which bacterial strains.

#### 4.3.1. Preparation

To perform the microbiological tests, 2 different mediums were prepared: PBS as a buffer solution to extract the antibiotics as an eluent from the test specimens and spacers, and Müller–Hinton Agar (MHA) to grow the bacterial colonies and perform the inhibition zone assays.

Specimens were placed in a separate tube with 20 mL of PBS.

The tubes were incubated at room temperature (25 °C) for 1 day, 7 days, 14 days, 28 days, and 42 days. At each time point, 2 mL of PBS eluent was removed and stored separately. The rest of the PBS was discarded, and the specimens were immersed in another 20 mL of fresh PBS. The tubes were then sealed again and placed upside down until the next extraction time point.

#### 4.3.2. Bacteria Preparation

To achieve bacterial suspensions for standardized microbial testing for all the inhibition zone assays, beads of each bacterial strain were plated and diluted on MHA in a Petri dish. These were incubated overnight at 37 °C.

A single colony or several colonies were removed with a swab and mixed into a saline solution (0.85% NaCl) until a McFarland Standard of 0.5 (±0.1) was achieved.

#### 4.3.3. Spacer Preparation and Eluate Extraction

Each spacer was placed in a beaker and submerged in 120 mL of PBS. The beakers were sealed with a sheet of aluminum and stored at room temperature (25 °C). At each time point (1 d, 7 d, 14 d, 28 d, and 42 d), 2 mL of eluent was removed and the beakers were refilled with 120 mL of fresh PBS.

#### 4.3.4. Inhibition Zone Test

A total of 60 µL of the eluent from each group for each time point was pipetted onto Petri dishes loaded with the specified bacteria. The Petri dishes were incubated overnight at 37 °C and the diameter of the inhibition zones was measured and documented in mm the next day. The rest of the eluent was frozen and preserved. The average diameter and standard deviation were calculated. Each cement concentration and time point was tested in triplicate.

## 5. Conclusions

PJI remains a challenging and increasingly common problem as the number of primary THAs and TKAs increases. If gentamicin-containing spacers must be used, Copal Exchange G is more effective than Cemex Spacer K against *S. aureus* and *E. coli*. The addition of high doses of vancomycin to a PMMA spacer based on double-containing PMMA cements such as Copal G + C and Copal G + V as part of a two-stage treatment for PJI can be an effective means of delivering local antibiotics, which appears to be one important variable for infection eradication. After the addition of vancomycin, the ISO compression strength is largely unchanged, the ISO bending modulus is increased, and ISO bending strength, DIN bending strength, and DIN impact, are reduced.

## Figures and Tables

**Figure 1 antibiotics-13-00818-f001:**
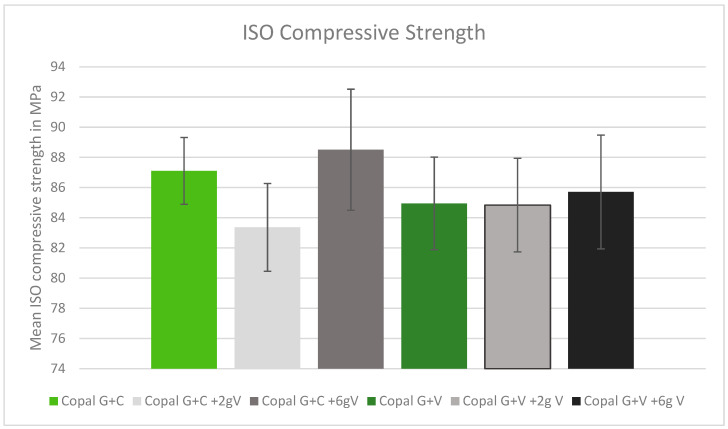
ISO compression strength in MPa with standard deviation. Controls: Copal G + C (light green) and Copal G + V (dark green) without manually added vancomycin. Adding 2 g vancomycin manually to Copal G + C in light grey; adding 6 g vancomycin manually in dark grey. Adding 2 g vancomycin manually to Copal G + V in grey, adding 6 g vancomycin in black. The ISO minimum was 70 MPa.

**Figure 2 antibiotics-13-00818-f002:**
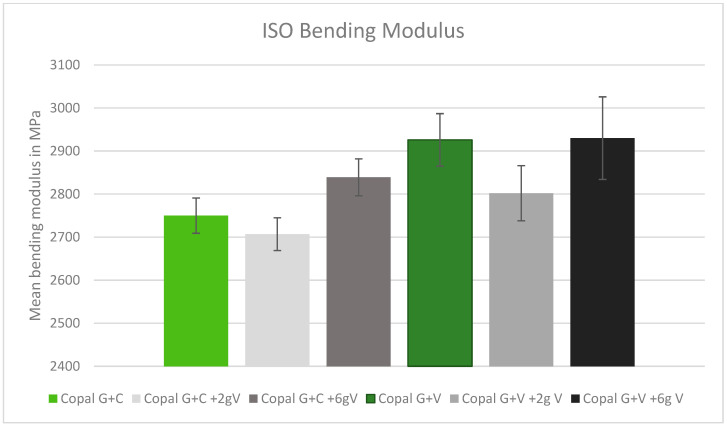
ISO bending modulus in MPa with standard deviation. Controls: Copal G + C (light green) and Copal G + V (dark green) without manually added vancomycin. Adding 2 g vancomycin manually to Copal G + C in light grey; adding 6 g vancomycin manually in dark grey. Adding 2 g vancomycin manually to Copal G + V in grey, adding 6g vancomycin in black. The ISO minimum was 1800 MPa.

**Figure 3 antibiotics-13-00818-f003:**
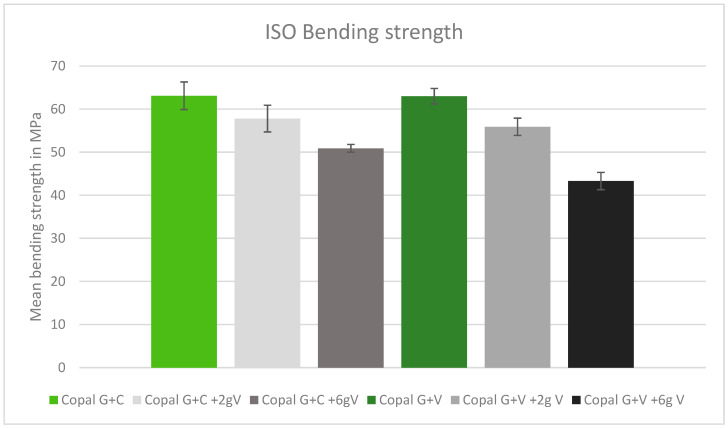
ISO bending strength in MPa with standard deviation. Controls: Copal G + C (light green) and Copal G + V (dark green) without manually added vancomycin. Adding 2 g vancomycin manually to Copal G + C in light grey; adding 6 g vancomycin manually in dark grey. Adding 2 g vancomycin manually to Copal G + V in grey, adding 6 g vancomycin in black. The ISO minimum was 50 MPa.

**Figure 4 antibiotics-13-00818-f004:**
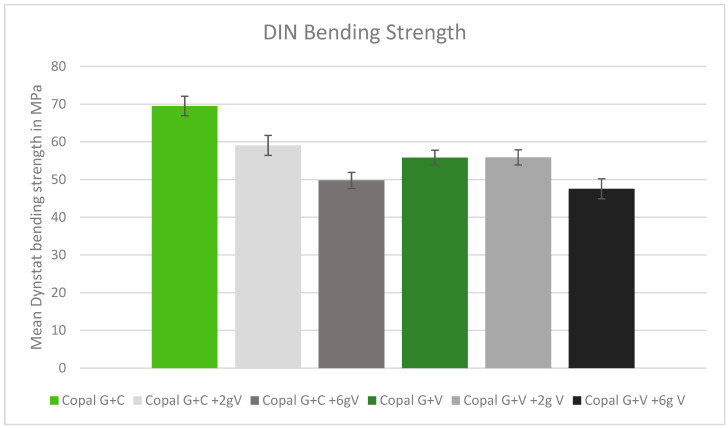
DIN bending strength in MPa with standard deviation. Controls: Copal G + C (light green) and Copal G + V (dark green) without manually added vancomycin. Adding 2 g vancomycin manually to Copal G + C in light grey; adding 6g vancomycin manually in dark grey. Adding 2 g vancomycin manually to Copal G + V in grey, adding 6 g vancomycin in black. The DIN minimum was 65 MPa.

**Figure 5 antibiotics-13-00818-f005:**
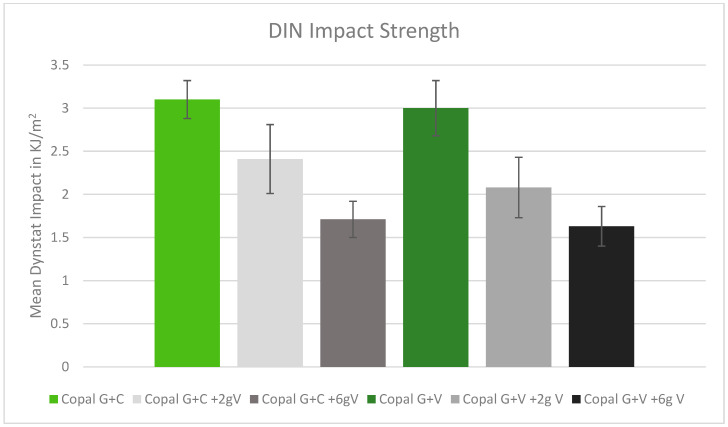
DIN impact strength in kJ/m^2^ with standard deviation. Controls: Copal G + C (light green) and Copal G + V (dark green) without manually added vancomycin. Adding 2 g vancomycin manually to Copal G + C in light grey; adding 6 g vancomycin manually in dark grey. Adding 2 g vancomycin manually to Copal G + V in grey, adding 6 g vancomycin in black. Strength compared with references.

**Figure 6 antibiotics-13-00818-f006:**
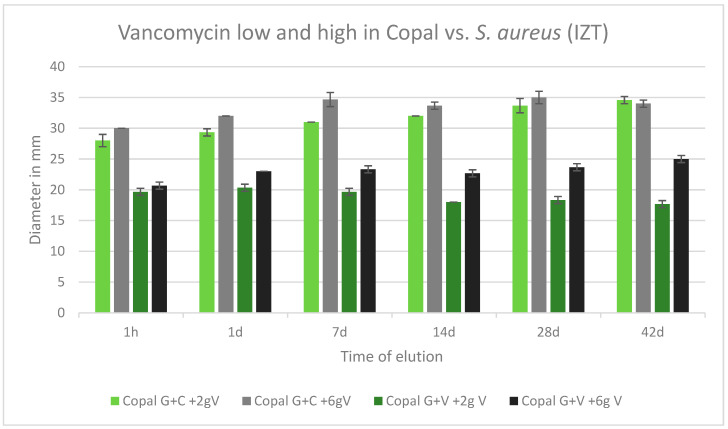
Inhibition zone test results against *S. aureus* in mm and standard deviation SD (*n* = 3) of Copal cements after adding low and high dosages of vancomycin over a period of 42 d. IZT inhibition zone test.

**Figure 7 antibiotics-13-00818-f007:**
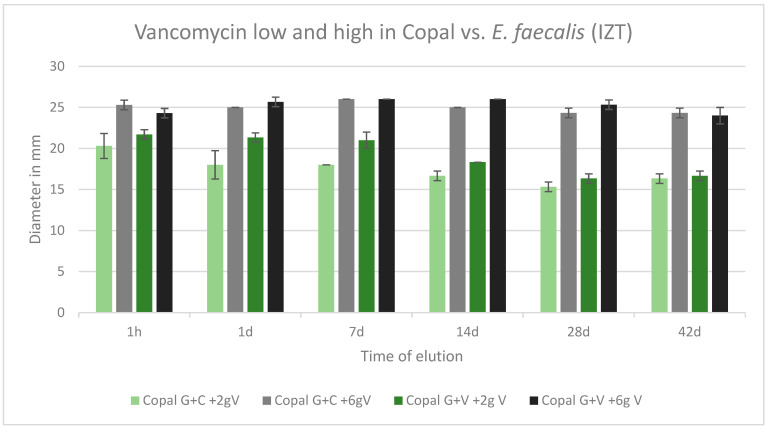
Inhibition zone test results against *E. faecalis* in mm and standard deviation SD (*n* = 3) of Copal cements after adding low and high dosages of vancomycin over a period of 42 d. IZT inhibition zone test.

**Figure 8 antibiotics-13-00818-f008:**
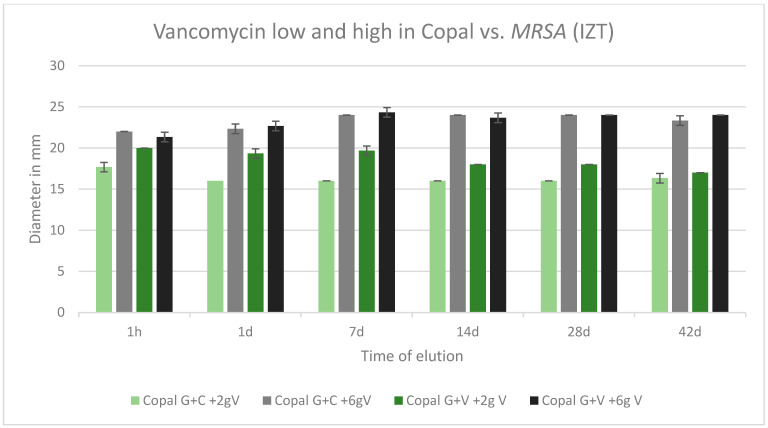
Inhibition zone test results against *MRSA* in mm and standard deviation SD (*n* = 3) of Copal cements after adding low and high dosages of vancomycin over a period of 42 d.

**Figure 9 antibiotics-13-00818-f009:**
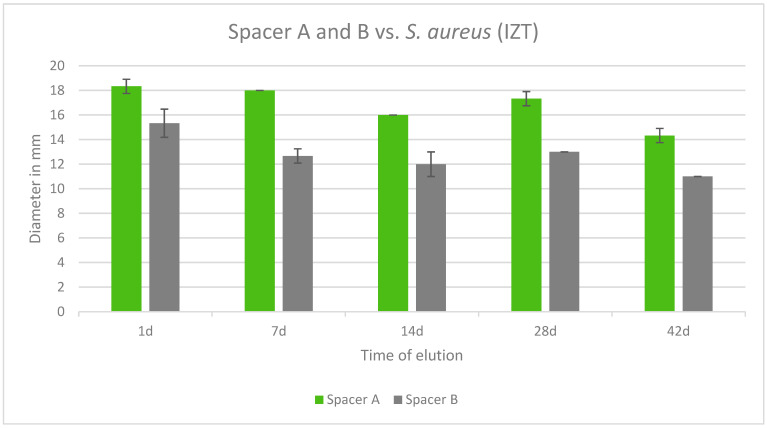
Inhibition zone test results in mm of Copal Exchange G and Cemex Spacer K against *S. aureus* over a period of 42 d (*n* = 3). ITZ inhibition zone test.

**Figure 10 antibiotics-13-00818-f010:**
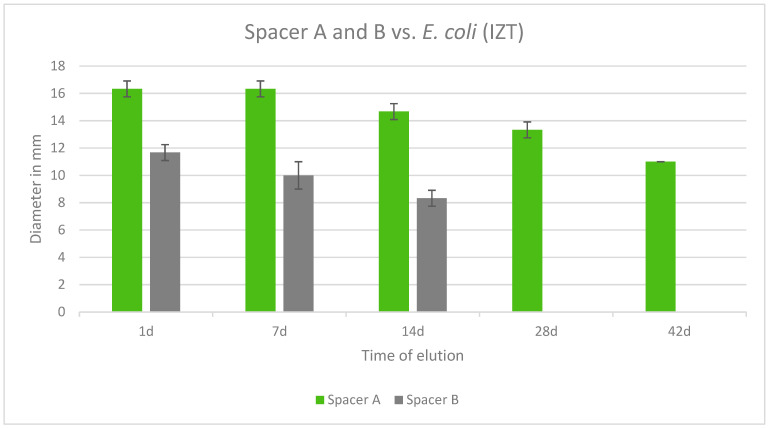
Inhibition zone test results in mm of Copal Exchange G and Cemex Spacer K against *E. coli* over a period of 42 d (*n* = 3). ITZ inhibition zone test.

**Figure 11 antibiotics-13-00818-f011:**
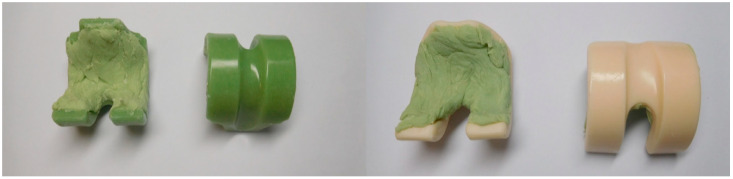
Femoral knee spacer remodeled with plain Palacos R on the non-articulating surface (back). **Left** = front and back of Copal Exchanged G (green), **Right** = front and back of Cemex Spacer K (crème colored).

**Table 1 antibiotics-13-00818-t001:** Overview of cements and preformed spacers used, the concentration of vancomycin added to the Copal cement powders, and the bacterial strains for microbiological testing. Copal spacer (Copal Exchange G), Tecres spacer (Cemex Spacer K). All cements and spacers contain industrially premixed gentamicin.

Cement/Spacer	Antibiotics	Tested Strains	
Copal G + C	2 g Vancomycin added manually	*S. aureus*	ATCC 29213
		*E. faecalis*	ATCC 29212
		*MRSA*	ATCC 43300
Copal G + C	6 g Vancomycin added manually	*S. aureus*	ATCC 29213
		*E. faecalis*	ATCC 29212
		*MRSA*	ATCC 43300
Copal G + V	2 g Vancomycin added manually	*S. aureus*	ATCC 29213
		*E. faecalis*	ATCC 29212
		*MRSA*	ATCC 43300
Copal G + V	6 g Vancomycin added manually	*S. aureus*	ATCC 29213
		*E. faecalis*	ATCC 29212
		*MRSA*	ATCC 43300
Copal Spacer		*S. aureus*	ATCC 29213
		*E. coli*	ATCC 25922
Tecres spacer		*S. aureus*	ATCC 29213
		*E. coli*	ATCC 25922

## Data Availability

All data are presented in this article. The data are also available at the Medical University of Graz (MUG).

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
