# Peer review of "High and Low Dosage of Vancomycin in Polymethylmethacrylate Cements: Efficacy and Mechanical Properties"

_antibiotics, 2024, doi:10.3390/antibiotics13090818_

Round 1

Reviewer 1 Report

Comments and Suggestions for Authors

This work is interesting to discuss the effect of Vancomycin on the efficacy and and mechanical properties of PMMA. The idea shows some novelty, however, some concerns should be well addressed to make this work more appealing:

1. There should be SEM images to depict the fracture surface area for comparsion.

2. In the conclusion part, some detailed data should be provided to depict how many doeages are high for be nephrotoxic.

3. There should be error bar to for some statistic analysis, for example, average compressive strength to represent at least 3-times measurement of the mechnical performance.

4. The reasearch writing is too redundant, the experimental sections seem to be mixed with discussion. Consequently, it is obsecure to understand the discussions and conclusions.

5. More materials characterizations such as FTIR etc, are suggested to provide.

Comments on the Quality of English Language

The English language of this work is moderate. The issue is on the writing, which is too wordy, without clear logical flow. I suggest that this paper shoud be rewrite to express his opinion shortly and clearly.

Reviewer 2 Report

Comments and Suggestions for Authors

1. One keyword illustrating the potential application of the study would help improve the visibility of the content

2. Some specifications on the differential composition of Copal C+G versus Copal G+V would be appreciated.

3. It would be appreciated to have information about the number of replications realized for each test (triplicate or less). Such information could be added in the figure caption.

4. The authors could specify the doses of vancomycin that can induce chondrotoxicity.

5. The authors should comment on the choice of the elution medium, namely PBS. The synovial fluid seems more appropriate for estimating the elution kinetics of vancomycin. Could the complex composition or the viscosity of the synovial fluid influence the elution characteristics of vancomycin?

6. Several paragraph are repeated:

 Line 401 – 404 and Line 426-429;

Line 391-393 and 417-418

7. The discussion is very rich, still, it is not centred on the obtained results. It would be important that the authors correlate more  the state of the art with their observation, with the use of vancomycin as antibiotic. This would give more credit and importance to their study.

 8. The obtained results should also be better underlined in the conclusion

Reviewer 3 Report

Comments and Suggestions for Authors

The manuscript has significant formatting issues.  section duplication, improper formatting of tables, grammatical & paragraph formatting issues, graph axis-labeling issues, etc.  The submitted version also has "comments" on.

There is no information regarding the strains of bacteria tested.

The tables are unlabeled regarding what is being presented, units, etc.

Overall the data indicates some interesting results, but it is impossible to effectively read and critically interpret the manuscript in it's current form.

Comments on the Quality of English Language

fine

Reviewer 4 Report

Comments and Suggestions for Authors

This paper on different dosages of antibiotics in PMMA is interesting, but too too long. Some parts can be cut and made "lighter" to read.

Some concepts on the different effects of different concentrations
of antibiotics on cement are well known and therefore these cocepts
can be concentrated and not continuosly repeated.
Moreover, is not clear the concentratiojn of clindamycin on Copal
Cement. Is not possible to study Copal alone without clndamycin
comparing Copal alone + vancomcin and Cemex spacer +
vancomycin?
The color utilsed to made graphs could be changed? These light
green is difficult to see.

Round 2

Reviewer 3 Report

Comments and Suggestions for Authors

The authors have made significant improvements to the manuscript.  There are several suggestions to improve:

1. The authors should clearly describe the total antibiotic formulation in table 1, specifically showing the commercially added vs. manually added antibiotics.  It will allow the readers to get a full understanding of which agents are in play, at which concentrations, and under which conditions.

2. The authors can consider condensing similar figures into a single figure with multiple panels.

3. The authors should state in the methods or discussion about the preparation of the cements.  Specifically, were the antibiotic efficacy tests done on cements that were previously used in the compression/bending tests?  One can envision a scenario where the pressure load on the cements may impact molecular integrity or the materials properties that would change release rates.

Comments on the Quality of English Language

fine
